# Effects of Voluntary Physical Exercise on the Neurovascular Unit in a Mouse Model of Alzheimer’s Disease

**DOI:** 10.3390/ijms241311134

**Published:** 2023-07-06

**Authors:** Jesús Andrade-Guerrero, Erika Orta-Salazar, Citlaltepetl Salinas-Lara, Carlos Sánchez-Garibay, Luis Daniel Rodríguez-Hernández, Isaac Vargas-Rodríguez, Nayeli Barron-Leon, Carlos Ledesma-Alonso, Sofía Diaz-Cintra, Luis O. Soto-Rojas

**Affiliations:** 1Laboratorio de Patogénesis Molecular, Laboratorio 4, Edificio A4, Carrera Médico Cirujano, Facultad de Estudios Superiores Iztacala, Universidad Nacional Autónoma de México, Tlalnepantla 54090, Mexico; jesusandrade1007@gmail.com (J.A.-G.); luis.d.rodriguez.h@comunidad.unam.mx (L.D.R.-H.); 2Departamento de Neurobiología del Desarrollo y Neurofisiología, Instituto de Neurobiología, Universidad Nacional Autónoma de México, Querétaro 76230, Mexico; erikaortabio@yahoo.com.mx (E.O.-S.); isaacvr.1108@gmail.com (I.V.-R.); barronleon1306@gmail.com (N.B.-L.); neurovet.cla@gmail.com (C.L.-A.); 3Red MEDICI, Carrera Médico Cirujano, Facultad de Estudios Superiores Iztacala, Universidad Nacional Autónoma de México, Tlalnepantla 54090, Mexico; citlalsalinas69@gmail.com; 4Departamento de Neuropatología, Instituto Nacional de Neurología y Neurocirugía Manuel Velasco Suárez, Ciudad de México 14269, Mexico; carlos.s.garibay@live.com.mx

**Keywords:** Alzheimer’s disease, physical exercise, neurovascular unit, pericytes, basal membrane, astrocytic end-feet, cerebral amyloid angiopathy, cognition

## Abstract

Alzheimer’s disease (AD) is the most common neurodegenerative disorder worldwide. Histopathologically, AD presents two pathognomonic hallmarks: (1) neurofibrillary tangles, characterized by intracellular deposits of hyperphosphorylated tau protein, and (2) extracellular amyloid deposits (amyloid plaques) in the brain vasculature (cerebral amyloid angiopathy; CAA). It has been proposed that vascular amyloid deposits could trigger neurovascular unit (NVU) dysfunction in AD. The NVU is composed primarily of astrocytic feet, endothelial cells, pericytes, and basement membrane. Although physical exercise is hypothesized to have beneficial effects against AD, it is unknown whether its positive effects extend to ameliorating CAA and improving the physiology of the NVU. We used the triple transgenic animal model for AD (3xTg-AD) at 13 months old and analyzed through behavioral and histological assays, the effect of voluntary physical exercise on cognitive functions, amyloid angiopathy, and the NVU. Our results show that 3xTg-AD mice develop vascular amyloid deposits which correlate with cognitive deficits and NVU alteration. Interestingly, the physical exercise regimen decreases amyloid angiopathy and correlates with an improvement in cognitive function as well as in the underlying integrity of the NVU components. Physical exercise could represent a key therapeutic approach in cerebral amyloid angiopathy and NVU stability in AD patients.

## 1. Introduction

Alzheimer’s disease (AD) is the most common type of dementia, affecting around 50 million people worldwide and it is estimated that this number will triple by the year 2050 [1], thereby becoming a global health priority. AD predominantly affects people 65 years of age or older and is characterized by affectation in cognition domains, such as memory, attention, orientation, language, and judgment, impacting the life quality of patients [2,3,4].

Histopathologically, AD is characterized by two hallmarks [5]: (1) intraneuronal aggregation of hyperphosphorylated tau protein; and (2) extracellular accumulation of amyloid beta (Aβ) peptide. The Aβ peptide is deposited in the form of neuritic plaques in the brain parenchyma but also in the walls of the cerebral blood vessels and the meninges, the latter leading to cerebral amyloid angiopathy (CAA). CAA is present in 70–97% of all cases of AD and causes cerebral microhemorrhages, cognitive impairment, changes in the morphology and structure of cerebral vasculature, and ultimately a neurovascular unit (NVU) dysfunction [6,7]. The NVU is composed of neurons, astrocytic end-feet, endothelial cells, pericytes, basement membrane, and extracellular matrix [8]. The pericytes are mural cells that are embedded in the basement membrane and extend their processes throughout the capillaries, participating in the stabilization of endothelial cells, regulation of the blood–brain barrier (BBB) permeability, angiogenesis, and clearance of the Aβ from the brain [9]. Astrocytes surround the capillaries and interact with endothelial cells through astrocytic end-feet, which can modulate the tone of adjacent vascular smooth cells, by triggering calcium signals that regulate the release of vasoactive mediators [10]. The vasoactive factors favor the flow of cerebrospinal fluid (CSF) toward the parenchyma and the elimination of solutes through the drainage of interstitial fluid. The basement membrane functions as a physical barrier that surrounds the abluminal portion of endothelial cells, providing stability, and mediates the anchoring and signaling of pericytes [11]. It has been suggested that NVU dysfunction leads to BBB breakdown, and a decrease in cerebral blood flow (CBF), which is associated with less nutrient and oxygen supply, impairment in cognitive domains, and acceleration of tau and Aβ pathology [12]. In this context, the vascular hypothesis of AD suggests that there is a neurovascular dysfunction that triggers the neuropathological hallmarks which then leads to neurodegeneration [7].

In this context, animal models are typically used to study the pathophysiology of neurodegenerative diseases like AD. These have been developed based on familiar mutations that express behavioral characteristics of the disease spectrum in human patients and show classical lesions replicating the symptoms of the disease [13]. The triple transgenic model for AD (3xTg-AD) develops cognitive deficits, Aβ plaques, and intracellular accumulation of the Tau protein [14]. Therefore, it represents a suitable model to understand the neuropathological mechanisms involved, as well as for the development of new therapeutic strategies. Non-pharmacological interventions, such as physical exercise, may attenuate disease risk or delay progression [15,16]. Growing evidence in patients with AD and animal models has shown the beneficial effects of exercise on brain function, proposing multiple cellular and molecular mechanisms: (i) release of neurotrophic factors, mainly brain-derived neurotrophic factor (BDNF) [17]; (ii) modulation of the neuroinflammation [18]; (iii) reduction of oxidative stress [19,20]; (iv) influencing the production of neurotransmitters, such as dopamine, serotonin, and norepinephrine [21,22]; and (v) promoting the maturation, proliferation, differentiation, and survival of neuronal cells [23]. Thus, modifiable lifestyle factors have received attention in efforts to combat AD.

Furthermore, in patients with mild cognitive impairment and AD, physical exercise helps improve global cognition, executive functions, and attention [24,25]. Likewise, most of the studies in AD transgenic mice have evidenced that aerobic exercise can alleviate learning and memory deficits and reduce tau and Aβ aggregation in the hippocampus [26,27]. Nonetheless, the effects of physical exercise in animal models for AD are variable, depending on the type of physical exercise implemented. Two types of physical exercise are used in rodents, “voluntary” and “forced” regimes, both of which have been shown beneficial effects on brain function [28,29]. The main advantages of voluntary exercise are less stress to the animal due to its manipulation by the researcher, natural satisfaction of innate game impulses, and facile use in long-term studies [28,29]. However, the effect of voluntary physical exercise on vascular amyloid deposits and NVU components in the 3xTg-AD mouse has not been evaluated. Therefore, this work aims to analyze the vascular amyloid deposits and their correlation with alteration in cognitive functions and NVU components and improvement after voluntary physical exercise in the 3xTg-AD mouse model. Here, we demonstrated that voluntary physical exercise in the 3xTg-AD promotes an improvement of cognitive behavior and decreases plaques and vascular amyloid deposits. Moreover, this effect correlates with a significant improvement in the morphology and structure of blood vessels, as well as in the different NVU components. Voluntary physical exercise could be proposed as a key strategy of non-pharmacological treatment to improve hippocampal-dependent cognitive ability and mitigate neurovascular alterations in the symptomatic stage of the disease.

## 2. Results

### 2.1. Voluntary Physical Exercise Does Not Cause Changes in Anthropometric Values and Physical Performance

No significant differences (*p* > 0.05) in body weight were observed between the 3xTg-AD and Non-Tg groups treated with voluntary physical exercise and the sedentary from the beginning to the end of the study (12 weeks) (Figure 1a). Interestingly, we observed that the 3xTg-AD exercise group has a lower body mass than the Non-Tg exercise group (*p* < 0.05) but only on weeks five (−7%), six (−5%), and eleven (−7%). Regarding gastrocnemius weight, we detected a significant decrease (*p* < 0.05) between Non-Tg and 3xTg-AD in both sedentary (−7%) and physical exercise groups (−4%). However, we observed a slight increase (*p* > 0.05; +4%) when we compared the 3xTg-AD sedentary to the 3xTg-AD exercise groups (Figure 1b). Additionally, a lower physical performance was found in the 3xTg-AD group compared to the Non-Tg group over 12 weeks of training (*p* < 0.05), and no differences were found over time between subjects in the same group (Figure 1c; *p* > 0.05).

### 2.2. Voluntary Physical Exercise Improves Learning and Memory

Animal learning and long-term memory were evaluated via the Barnes Maze. In the training phase, no statistical differences (*p* > 0.05) were found between different groups over four days (Figure 2a). In the evaluation phase, we observed a significant increase (+202%; *p* < 0.05) in latency retention time in the 3xTg-AD sedentary group when compared with the Non-Tg group (Figure 2b). Nonetheless, voluntary physical exercise decreased slightly this parameter (*p* > 0.05; −36%) in the 3xTg-AD group, and no statistical differences (*p* > 0.05) were found between the Non-Tg and 3xTg-AD groups (Figure 2b). Likewise, a significant decrease was noticed in quadrant latency time in the 3xTg-AD sedentary (−51%; *p* < 0.05) and exercise (−17%; *p* < 0.05) groups when compared with the Non-Tg group (Figure 2c). Also, there was a non-significant increase in this parameter (+68%; *p* > 0.05) in the 3xTg-AD group after exercise intervention (Figure 2c). Finally, we observed a higher number of errors in the 3xTg-AD sedentary compared with the Non-Tg group (+262%; *p* < 0.05), but not after the voluntary physical exercise (*p* > 0.05; Figure 2d).

### 2.3. Voluntary Physical Exercise Decreased Amyloid Deposits in the Hippocampus of 3xTg-AD Mice

As we expected, a significant decrease (*p* < 0.05; −56%) in amyloid deposits was observed in the hippocampus subiculum of the 3xTg-AD exercise group compared to the sedentary group. These amyloid deposits were not observed in the Non-Tg group (Figure 3).

### 2.4. Exercise Caused Changes in Cerebral Vessels in the Hippocampus of 3xTg-AD Mice

Exercise causes different cerebral vascular changes (Figure 4). In the hippocampal fissure, a significant increase was observed in the area (*p* < 0.05; +91%) and perimeter of blood vessels when the Non-Tg and 3xTg-AD groups were compared (*p* < 0.05; +44%; Figure 4b,c). The Feret diameter was significantly larger in the sedentary 3xTg-AD group compared to the Non-Tg group (*p* < 0.05; +56%; Figure 4d). All these measures decreased after treatment with voluntary physical exercise (*p* < 0.05; −19% for the area; −2% for perimeter; −13% for Feret diameter; Figure 4a–d). Interestingly, we observed a non-significant increase in the perivascular space (PVS) of the 3xTg-AD sedentary group (*p* > 0.05; +78%) and the exercise group (*p* > 0.05; +8%) compared with the Non-Tg sedentary group. This PVS decreased (*p* > 0.05; −39%) in the 3xTg-AD group with voluntary physical exercise compared to the sedentary 3xTg-AD group (Figure 4e). Moreover, we identified an increase in the ratio between the number of vessels, with and without PVS, in the 3xTg-AD sedentary compared to the Non-Tg group (*p* < 0.05; +126%). This ratio decreased in the 3xTg-AD sedentary compared with the 3xTg-AD exercised group (*p* < 0.05; −69%; Figure 4f). Finally, we found vascular amyloid deposits in the 3xTg-AD mice, which significantly decreased after the exercise therapy (*p* < 0.05; −35%; Figure 4g).

### 2.5. Voluntary Physical Exercise Produced Changes in the Neurovascular Unit in the Hippocampus

Astrocytic end-feet are cellular components of the NVU. Astrocytic end-feet were evaluated using the aquaporin-4 (AQ4) antibody. In the hippocampal fissure, we observed a significantly greater area of immunostaining against AQ4 (*p* < 0.05; +72%) in the 3xTg-AD group compared to the Non-Tg group in sedentary conditions. Moreover, we detected a modest decrease after voluntary physical exercise intervention in the 3xTg-AD group (*p* > 0.05; −21%; Figure 5).

Other cellular components of NVU are the pericytes, which were detected via the PDGFR-β marker antibody. We found that exercise caused a significant increase (*p* < 0.05; +33%) in the PDGFR-β immunofluorescence (IF) area of the 3xTg-AD group compared with the sedentary group. Moreover, we observed a decrease of −22% (*p* > 0.05) when the 3xTg-AD was compared with both Non-Tg groups in sedentary condition (Figure 6a,b). On the other hand, the basement membrane is a key structure of NVU, which herein was evaluated using the collagen IV antibody. We found a significant decrease in the 3xTg-AD sedentary group compared to the Non-Tg group (*p* < 0.05; −42%). Interestingly, the exercise intervention triggered a significant increase (*p* < 0.05; −17%) in the 3xTg-AD group with exercise compared to the sedentary condition (Figure 6a,c).

### 2.6. The Amyloid Deposits Correlate with NVU Alteration and Cognitive Deficits in Sedentary 3xTg-AD and Improvement after Voluntary Exercise Intervention

Vascular amyloid deposits correlate with the alteration of the NVU components in 3xTg-AD and an improvement after voluntary exercise intervention (Figure 7): collagen IV (basement membrane; R^2^ = 0.5823, *p* = 0.0168; panel a), PDGFR-β (pericytes; R^2^ = 0.5068, *p* = 0.0476; panel b), AQ4 (astrocytic end-feet; R^2^ = 0.6351, *p* = 0.0179; panel c) and PVS (R^2^ = 0.8127, *p* = 0.0022; panel d). In addition, the cognitive deficit (latency retention time) is correlated with both amyloid vascular aggregates (R^2^ = 0.6147, *p* = 0.0073; panel e) and amyloid plaques (R^2^ = 0.6103, *p* = 0.0129; panel f).

## 3. Discussion

Our results demonstrate that in symptomatic stages of the 3xTg-AD model, there are vascular amyloid deposits in the hippocampus, which correlated with cognitive dysfunction and NVU alterations: (1) changes in vascular morphology (loss in circularity and increase in the perimeter and area) and increase in the PVS; (2) increase in immunodetection of the astrocytic end-feet; and (3) decrease in pericyte and basement membrane immunoreactivity. In addition, we showed that the intervention with voluntary physical exercise in this model triggers an improvement in the vascular amyloid deposits, NVU components and the cognitive functions.

CAA worsens AD pathology through a decreased Aβ clearance and dysfunction in perivascular drainage, which is associated with cognitive impairment [30,31]. Likewise, in the present study, we showed hippocampal vascular amyloid deposits that correlated with cognitive deficit, which improved after physical exercise in 3xTg-AD mice. The growing evidence suggests that aerobic exercise ameliorates learning and memory deficits, and attenuates the amyloid burden in mice models of AD [26,32]. On the other hand, exercise training increases hippocampal volume, and reduces age-related decline in memory function in humans [33].

The mechanisms by which physical exercise reduces amyloid angiopathy have not been elucidated. We hypothesize that physical exercise could have beneficial effects at the neurovascular level through the following mechanisms: (1) protective effects on NVU, including neurons, astrocytes, pericytes, and the extracellular matrix [34]; (2) promoting Aβ clearance through increase or enhancement of low-density lipoprotein receptor-related protein 1 (LRP1), neprilysin, matrix metalloproteinase-9 (MMP9), and insulin-degrading enzyme (IDE) [35,36]; (3) improvement of the glymphatic system [37]; and (4) stabilizing the BBB and therefore the permeability to toxic molecules or proinflammatory cells [38].

The vascular Aβ deposition has been associated with NVU components degeneration and dysfunction: endothelial cell [39,40], astrocytic end-feet [7], pericytes [7,41], and smooth muscle [7,42], and enlarged PVS [43]. The PVS, also known as Virchow–Robin space, is composed of fluid compartments surrounding the cerebral small blood vessels and conduit the fluid transport, permitting exchange between cerebrospinal fluid and interstitial fluid, and clearance of toxic products from the brain [44]. Different PVS metrics, including number, diameter, and volume, are quantified via structural and harmonized MRI techniques [44,45]. Therefore, enlarged PVS has been proposed as an early imaging biomarker for AD [43]. Both in AD and other dementias, CAA, and various vascular pathologies [45,46], the PVS could be dysfunctional causing an alteration in interstitial fluid and metabolite homeostasis [46,47]. In addition, neuroinflammation is associated with enlarged PVS, and this could trigger the breakdown of the extracellular matrix and affect the BBB integrity [48]. In our study, we found an increase in PVS measurement, which correlated with vascular amyloid deposits. Moreover, the histological analysis evidenced that physical exercise decreased both vascular amyloid deposits and PVS. It is not clear how physical exercise decreases PVS, however, we suggest an improvement in the efficiency of the glymphatic system by preserving the perivascular polarization of AQP4 channels and promoting Aβ clearance [37].

Astrocytic end-feet express AQ4, a water channel protein, involved in water homeostasis and vasogenic edema formation. Recent studies began to show possible interrelationships between AQ4 and neuroinflammation [49]. Upregulation of AQP4 has been associated with early edema, BBB disruption, and increased secretion of proinflammatory cytokines [50]. Here, we found an increase in AQ4 immunoreactivity in 3xTg-AD mice, suggesting that at this stage, the model could be in a resolution phase of vasogenic edema [50]. However, the exact role of this increase in AQP4 is still under discussion.

On the other hand, it has been suggested that the basement membrane is thickening and is associated with amyloid deposits in AD patients [51]. This pathological finding in turn affects the functioning and stability of the BBB and the contact between endothelial cells and pericytes [52]. This basement membrane thickening in the temporal cortex, detected via collagen IV, is not related to an increase in vessel density [53]. Collagen IV could protect the neurons against Aβ, blocking the association with the neuron [54]. One study demonstrated an increase in collagen IV in the cortex of 11-month-old 3xTg-AD mice [55]. The thickening of the basement membrane could be due to the imbalance between the synthesis and degradation of the proteins that constitute it [56]. Here, we found a decrease in collagen IV immunoreactivity in the hippocampus at 13 months of age in the 3xTg-AD model, which could suggest basement membrane remodeling as a pathological process [56]. Moreover, this thinning in the basement membrane could be secondary to the action of matrix metalloproteinase (MMP-2 and 9) that degrade basal lamina [57]. Also, these two molecules may, in turn, be associated with vascular amyloid deposits and BBB rupture [58]. Furthermore, MMP-9 aggravates the development of vasogenic edema by degradation of the basal lamina located between the astrocytic end-feet and the endothelium [59]. In this study, the decrease in collagen IV immunoreactivity was correlated with vascular amyloid deposits, returning to basal levels after physical exercise. The mechanisms through which physical exercise exerts its beneficial effects on the basement membrane are not clear, however, it has been suggested that it increases the expression of collagen IV and MMP-9, which plays a key role in exercise-strengthened collagen IV expression against ischemia or reperfusion injury [60].

Finally, postmortem studies of AD patients have shown a loss of pericytes that are associated with an increase in proinflammatory molecules and MMP-9, which culminates in a BBB breakdown [61]. In addition, this loss of pericytes has been associated with increased brain levels of Aβ40 and Aβ42, accelerating amyloid angiopathy, the development of tau pathology, and early neuronal loss, thereby resulting in cognitive impairment [9]. Similarly, we observed a decrease in pericytes in the 3xTg-AD model associated with cerebral amyloid deposits and an increase in these cells after the intervention with voluntary physical exercise. It is not clear how exercise effects pericytes, however, it has been suggested that it might improve pericyte proliferation and that it upregulates PDGFRβ in transgenic mice with AD [62].

Some limitations of this study were that it did not explore the effect in males, nor did it study the effect of the intervention with forced physical exercise. It also did not explore whether there is an alteration in the BBB. Therefore, future research should evaluate the effects of forced physical exercise on the NVU, discern if there is any sexual dimorphism, and analyze the changes in the BBB permeability, as well as the morphostructural changes in the NVU through electron microscopy.

In conclusion, our results showed vascular amyloid deposits which correlated with NVU alteration and cognitive dysfunction in the 3xTg-AD model. Moreover, we found that voluntary physical exercise decreases vascular amyloid deposits and improves NVU components and cognitive behavior. Thus, voluntary physical exercise could represent a non-pharmacological therapeutic strategy to decrease AAC and stabilize the components of NVU in AD patients.

## 4. Materials and Methods

### 4.1. Animals

This study was experimental and focused on behavioral and histological analysis. The triple transgenic model for Alzheimer’s disease (3xTg-AD) was used, which harbors the APPSwe and tauP301L transgenes in knock-in of PS1M146V and shows intracellular and extracellular amyloid-β (Aβ) and tau pathology dependent on age [14]. All mice included in the study were females around 10 months of age (symptomatic stage) since the pathology is exacerbated in them [63]. Twenty 3xTg-AD and twenty non-transgenic (Non-Tg; 129C57/BL6) mice were genotyped and randomly assigned to exercise or sedentary groups (Appendix A). They were housed in groups of 4–5 per cage of the same genotype with access to food and water ad libitum under standard conditions (12:12 h light–dark cycle, at room temperature of 22 ± 2 °C and relative humidity of 60 ± 5%). Animal handling was supervised by a veterinarian. One week before beginning the behavioral tests or the application of physical exercise, the animals were habituated to the researcher. This study was licensed in accordance with the principles established in the NIH guide for the care and use of laboratory animals and was approved by the Bioethics Committee of the Institute of Neurobiology of the National Autonomous University of Mexico with protocol number 117.

### 4.2. Voluntary Physical Exercise Regimen

A scheme was designed for the mice assigned to the voluntary exercise groups, using an exercise wheel (Panlab LE905, Barcelona, Spain), for 12 weeks (at the end the animals were 13 months old), with a frequency of 5 times per week where the mice had free access to the wheel for 2 h (Appendix A). The movement of the spinning wheel was detected via a sensor mounted on the top of the cage, which was transmitted to a PC controlled by an internal program to measure the running distance (kilometers) of the individual animal per day. The body weight was measured over 12 weeks in the two groups, 3xTg-AD and Non-Tg, both under sedentary and exercise conditions. At the end of the study, the gastrocnemius was dissected and weighed through an analytical plus scale (Ohaus, Parsippany, NJ, USA).

### 4.3. Barnes Maze

Spatial memory and learning were evaluated using the Barnes maze as described elsewhere [64]. The maze consists of a raised circular platform with 20 holes spaced evenly around the perimeter. An escape box is mounted under one hole, while the remaining 19 holes are left empty. Both bright light and open spaces are aversive to rodents, thus factors inducing an escape behavior. The test has two phases; first, the training latency phase over 4 days, where the rodents were placed in the platform for 3 min and the time that the mouse took to reach the escape box was measured. Next is the retention phase, wherein 24 h later, the escape box was removed, and the mice were placed again in the platform and we evaluated the time that the mouse took to reach where the escape box was (latency retention); the time spent in the respective quadrant (quadrant latency) and the number of errors made were also evaluated. All these parameters were videotaped (Smart Video Tracking software 2.5, Panlab, Barcelona, Spain) for 90 s.

### 4.4. Histological Analysis

Animals were anesthetized with intraperitoneal (i.p.) pentobarbital lethal dose (300 μg/kg body weight) and euthanized by transcardial perfusion of 4% paraformaldehyde (PFA) in phosphate-buffered saline solution (PBS) with 1000 IU of heparin and 0.1% procaine. Brains were removed and post-fixed for 24 h in a buffered 4% PFA solution at room temperature. Subsequently, the brains were embedded in paraffin and serial coronal sections were conducted (5 μm thickness, 2.46–2.70 mm ML from Bregma according to the Paxinos atlas) using a sliding microtome (Leica RM 2135, Wetzlar, Germany) to obtain the hippocampal region.

#### 4.4.1. Immunohistochemistry Assays

The tissues were deparaffinized at 60°, hydrated, and incubated with 1% hydrogen peroxide; epitope unmasking was carried out in a citrate buffer for 10 min in a microwave and later were incubated with a blocking 0.2% bovine serum albumin solution (Sigma-Aldrich, St. Louis, MO, USA) for 1 h. Subsequently, the respective primary antibodies anti-mouse Aβ (BAM-10, 1:500, Sigma-Aldrich, St. Louis, MO, USA) and mouse IgG monoclonal AQP4 (1:30; Santa Cruz Biotechnology; Dallas, TX, USA) were incubated in a humid chamber at 4 °C overnight. On the next day, the tissues were incubated with the corresponding secondary biotinylated antibodies (1:500, Invitrogen Molecular Probes; Eugene, OR, USA) for 2 h at room temperature, and revealed using the ABC/diaminobenzidine kit. Finally, the tissues were counterstained with hematoxylin-eosin (to mark the entire cell both nucleus and cytoplasm) or only hematoxylin, and later they were mounted on slides using Entellan resin (Merck, KGaA; Darmstadt, Germany).

Digital images were captured using a light microscope (Eclipse Ci-Li; Nikon, Japan) with a Nikon Plan 20×/0.40 air lens. The percentage of immunoreactivity area (μm/cm^2^) of AQ-4 and Aβ deposits in both the subiculum (plaques) and hippocampal fissure (blood vessels) were analyzed. Additionally, we evaluated the vascular morphology based on the following parameters: (1) vessel area and perimeter, (2) Feret diameter (the longest distance between any two points along the selection boundary), (3) area of perivascular space, and (4) vessel ratio (vessels with perivascular space divided by vessels without this space). Both immunoreactivity and vascular morphology were evaluated in three random photomicrographs in four-five anatomic levels per animal using the Image J software version 8 (National Institutes of Health, Bethesda, MD, USA).

#### 4.4.2. Double Immunofluorescence for Pericytes and Basal Membrane

For double immunofluorescence assays, we followed the same steps as immunohistochemistry assays, only adding pre-treatment with 0.4% proteinase K solution for 10 min. Subsequently, the samples were incubated at 4 °C overnight with the following pair of primary antibodies: anti-rabbit-PDGFR β to detect pericytes (1:250 Abcam, Cambridge, UK), and anti-mouse-Collagen IV to the basal membrane (1:300, Sigma Aldrich, St. Louis, MO, USA). After incubation, the tissues were washed and incubated with a suitable cocktail of secondary antibodies for 1 h at room temperature: Alexa Fluor 488 goat anti-rabbit H+L IgG (1:300; Invitrogen Molecular Probes; Eugene, OR, USA), and Alexa Fluor 594 donkey anti-mouse H+L IgG (1:300; Invitrogen Molecular Probes; Eugene, OR, USA). The nuclei were counterstained with 1 μM DAPI (Sigma-Aldrich; St. Louis, MO, USA), and the slices were mounted on glass slides using VECTASHIELD (Vector Laboratories; Burlingame, CA, USA). Fluorescence images were obtained via a Zeiss, Axio Vert. A1 FL microscope (Zeiss; Berlin, Germany) with 385 nm (UV), 555 nm (green fluorescence), and 630 nm (red fluorescence) filters using a 63× objective. The images were digitized with an Axiocam 208 color camera; later, they were exported to the Image J software version 8 (National Institutes of Health, Bethesda, MD, USA), and then the immunofluorescence area of blood vessels was measured, as described in the previous subsection.

### 4.5. Statistical Analysis

All values were normalized, and data were presented as the mean value ± the standard error of the mean (S.E.M.). Data were analyzed using *t* student and two-way analysis of variance (ANOVA) and pairwise comparisons between groups, performed using Tukey’s post hoc test analysis. Finally, the results were processed, and the construction of the graph was carried out with the PRISMA 8 software, where the statistical significance was considered at least *p* ≤ 0.05. For correlation analysis, only sedentary and exercise conditions of 3xTg-AD were evaluated, and then Pearson’s correlation coefficient and subsequent linear regression were determined. The statistical difference was considered at *p* < 0.05.

## Figures and Tables

**Figure 1 ijms-24-11134-f001:**
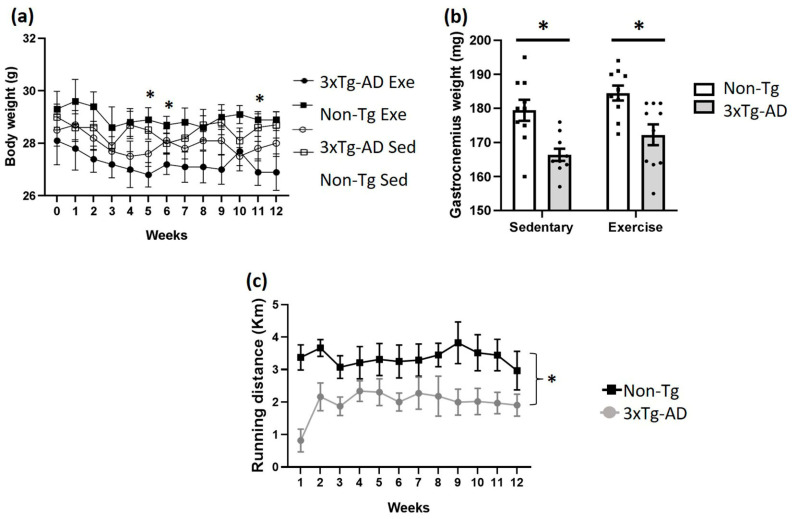
Effects of voluntary physical exercise on anthropometric values and physical performance in 3xTg-AD mice. (**a**) Body weight was measured over 12 weeks. Repeated measures ANOVA, Tukey’s post hoc, values are means ± SEM (*n* = 10 per group), * *p* < 0.05 when compared 3xTg-AD exercise group vs. Non-Tg exercise group. In panel (**b**), the graph shows gastrocnemius weight after 12 weeks of voluntary physical exercise. Two-ways ANOVA, Tukey’s post hoc, values are means ± SEM (*n* = 10 per group), * *p* < 0.05 when compared 3xTg-AD group vs. Non-Tg group. (**c**) Distance traveled on wheels (km) over 12 weeks. Repeated measures ANOVA, Tukey’s post hoc, values are means ± SEM (*n* = 10 per group), * *p* < 0.05. Abbreviations: Exe, exercise; Sed, sedentary.

**Figure 2 ijms-24-11134-f002:**
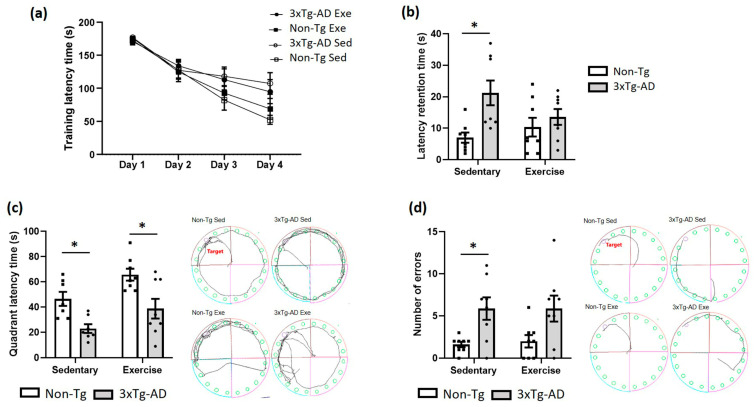
Effects of voluntary physical exercise on learning and long-term memory via the Barnes Maze. (**a**) Training latency phase over 4 days. Repeated measures ANOVA, Tukey post hoc, values are means ± SEM (*n* = 8–9 per group). (**b**) The graph shows the retention time evaluated 24 h after the training phase. The time in the quadrant (**c**) and the number of errors (**d**) are shown in the graphs and representative schemes. Two-way ANOVA, Tukey’s post hoc, values are means ± SEM (*n* = 8–9 per group), * *p* < 0.05. Abbreviations: Exe, exercise; Sed, sedentary.

**Figure 3 ijms-24-11134-f003:**
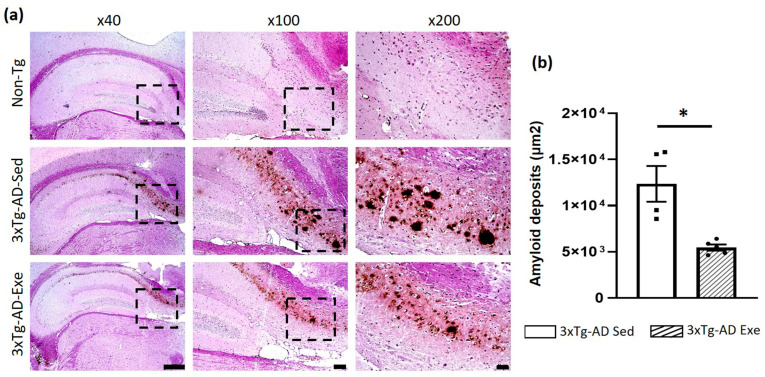
Effect of voluntary physical exercise on Aβ deposition in the hippocampus of the mice. (**a**) Representative photomicrographs of amyloid deposits of the hippocampal subiculum. Different magnifications (the black dashed box) are illustrated. Scale bar = 500 μm for ×40 magnification; 100 μm for ×100; and 50 μm for ×200. (**b**) The graph shows the effect of voluntary physical exercise on hippocampal amyloid beta deposits. T student, values are means ± SEM (*n* = 5 per group), * *p* < 0.05. Abbreviations: Exe, exercise; Sed, sedentary.

**Figure 4 ijms-24-11134-f004:**
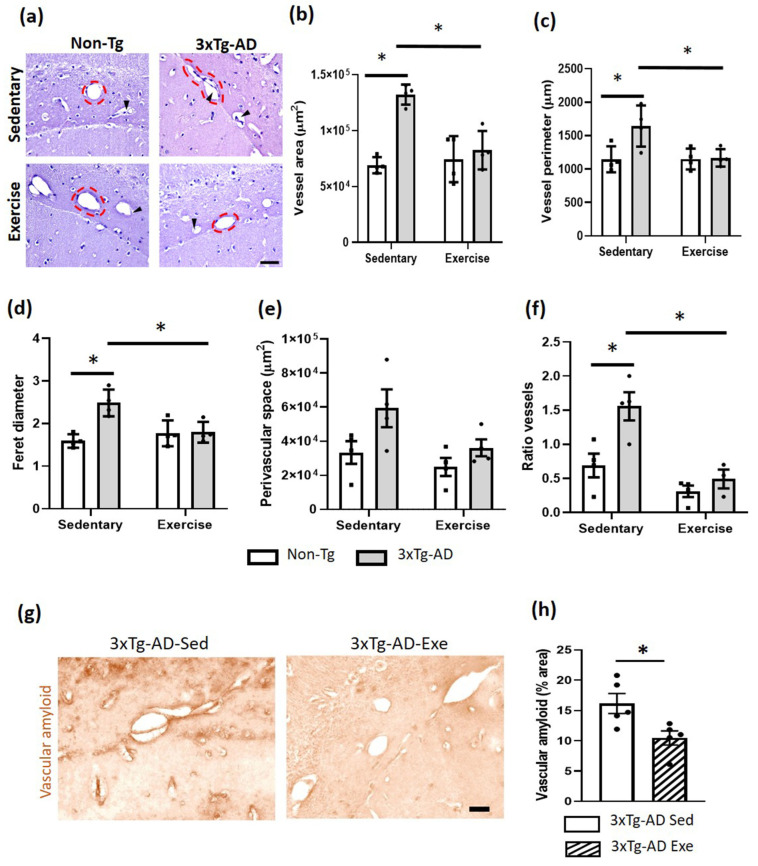
Effects of the voluntary physical exercise on the brain vasculature in the hippocampal fissure of 3xTg-AD mice. (**a**) Representative photomicrographs of hippocampal fissure blood vessels stained with hematoxylin-eosin under different conditions. The dashed circles show the different vascular morphology between groups and the black arrowheads point to the perivascular space. Scale bar = 20 μm common to all micrographs. The graphs show the area (**b**), perimeter (**c**), Feret diameter (**d**), perivascular space (**e**), and the ratio of vessels (number of vessels with and without perivascular space; panel (**f**). Two-way ANOVA, post hoc Tukey, mean ± SEM measured at four anatomical levels (*n* = 4–5 per group), * *p* < 0.05. Panel (**g**) illustrates representative photomicrographs of the vascular amyloid deposits. Scale bar = 20 μm common to all micrographs. (**h**) The graph shows the measurement of the percentage of area occupied by amyloid vascular deposits. T student, values are mean ± SEM calculated from four anatomical levels (*n* = 4–5 per group). * *p* < 0.05. Abbreviations: Exe, exercise; Sed, sedentary.

**Figure 5 ijms-24-11134-f005:**
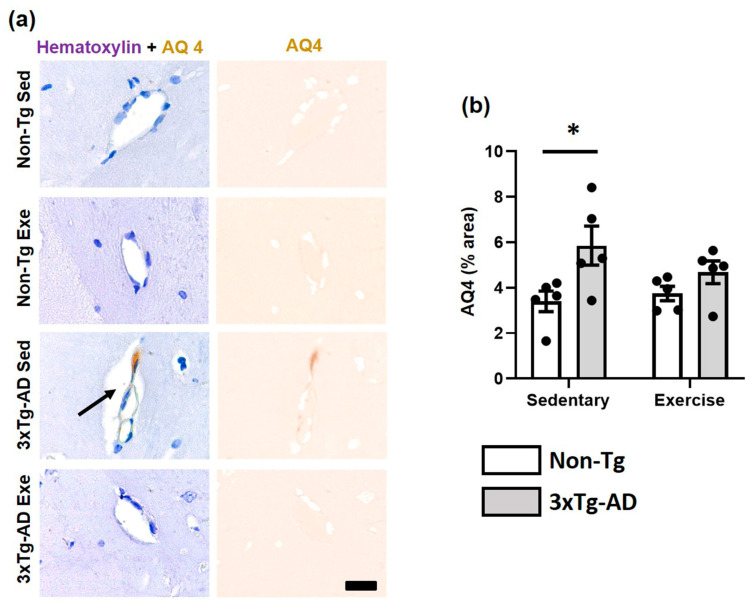
Effect of voluntary physical exercise on astrocytic end-feet. (**a**) Representative photomicrographs of AQ4 (astrocytic end-feet) in hippocampal fissure blood vessels. Scale bar = 20 μm common to all micrographs. The black arrow points out the prominent perivascular space. (**b**) Graphs show astrocyte end-feet area. Two-way ANOVA, Tukey post hoc, mean ± SEM measured at four anatomical levels (*n* = 5 per group), * *p* < 0.05. Abbreviations: AQ4, aquaporin-4; Exe, exercise; Sed, sedentary.

**Figure 6 ijms-24-11134-f006:**
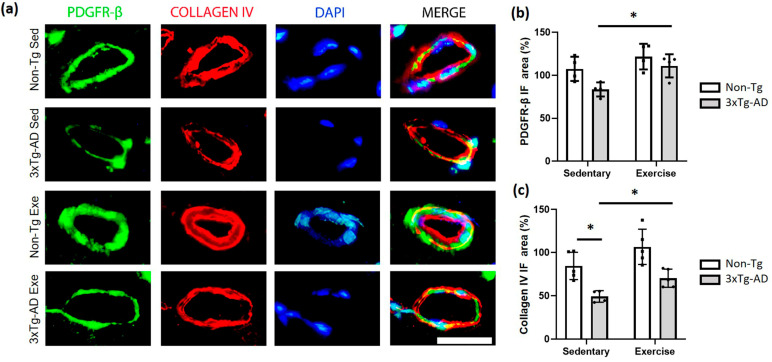
Effect of voluntary physical exercise on pericytes and basement membrane in the hippocampus. (**a**) Fluorescent photomicrographs of pericyte (PDGFR-β; green channel) and basement membrane (collagen IV; red channel) in hippocampal fissure blood vessels. The cellular nuclei were stained with DAPI (blue channel). Scale bar = 20 μm common to all micrographs. The graphs show the area of the pericyte (**b**) and basement membrane (**c**). Two-way ANOVA, post hoc Tukey, mean ± SEM measured at four anatomical levels (*n* = 4–5 per group), * *p* < 0.05. Abbreviations: Exe, exercise; IF, immunofluorescence; Sed, sedentary.

**Figure 7 ijms-24-11134-f007:**
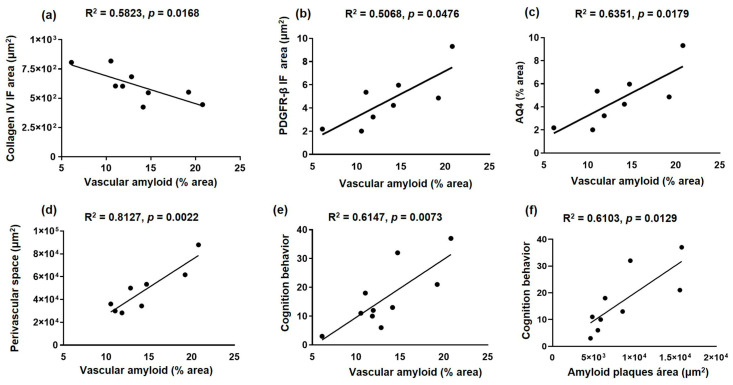
Correlation between amyloid deposits and neurovascular unit and cognition alteration in sedentary and exercised 3xTg-AD mice. The graphs show the correlation between vascular amyloid deposits and collagen IV (basal membrane) immunofluorescence (IF) area (**a**), PDGFR-β (pericytes) IF area (**b**), AQ4 area (astrocytic end-feet) (**c**), perivascular space (**d**), and cognition behavior (latency retention time) (**e**), such as between amyloid plaques and cognition behavior (**f**), in the hippocampal fissure. Pearson’s correlation coefficient and linear regression appear at the top of every graph *p* < 0.05 was considered a statistically significant difference.

## Data Availability

The data that support the findings of this study are available from the corresponding author upon reasonable request.

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
