# Peer review of "Effects of Voluntary Physical Exercise on the Neurovascular Unit in a Mouse Model of Alzheimer’s Disease"

_ijms, 2023, doi:10.3390/ijms241311134_

Round 1

Reviewer 1 Report

Thank you for the opportunity to read this well written, clear and concise study. It was interesting and a pleasure to read, although the following modifications are proposed:

The type of study carried out must appear in the title of the work.

The introduction is very complete and allows the reader to know the main topic of the research, informs about the purpose and importance of the work in the clinical field, and also answers the question posed in the scientific context. It includes previous works on the subject in question and makes clear the detailed aspects of the review, which constitutes the object of the proposed investigation. It explains the general problem of the research, includes previous work on the topic in question, and specifies the objective of the study.

The authors of this article have meticulously explained the results by providing relevant tables to what is explained in the text. On the other hand, they have provided a clear and complete discussion comparing their results with previous studies and arguing the existing differences. However, the limitations of this research are not presented. Please add.

The "methods" section is one of the most fundamental sections of a scientific article with these characteristics and is well developed and organized. However, a first subsection must be added with the information referring to the type of study and the code of the favorable certificate of the ethics committee. Also, a flowchart should be added.

Author Response

Reviewer 1

Thank you for the opportunity to read this well written, clear and concise study. It was interesting and a pleasure to read, although the following modifications are proposed:

  • We appreciate the reviewer for analyzing our manuscript and the suggestions made to improve it. The changes have been highlighted in yellow.

Q1. The type of study carried out must appear in the title of the work.

A1. We appreciate the reviewer's suggestion, however, throughout the manuscript, we mention that the type of study was histological and behavioral (please see L31, L282, and L339). Therefore, we consider that it is not necessary to add it to the title.

  • The introduction is very complete and allows the reader to know the main topic of the research, informs about the purpose and importance of the work in the clinical field, and also answers the question posed in the scientific context. It includes previous works on the subject in question and makes clear the detailed aspects of the review, which constitutes the object of the proposed investigation. It explains the general problem of the research, includes previous work on the topic in question, and specifies the objective of the study.
  • We thank the reviewer´s comments.

Q2. The authors of this article have meticulously explained the results by providing relevant tables to what is explained in the text. On the other hand, they have provided a clear and complete discussion comparing their results with previous studies and arguing the existing differences. However, the limitations of this research are not presented. Please add.

A2. We appreciate the reviewer's suggestion and have added the limitations and perspectives of this research (please see L326 to L330).

Q3. The "methods" section is one of the most fundamental sections of a scientific article with these characteristics and is well developed and organized. However, a first subsection must be added with the information referring to the type of study and the code of the favorable certificate of the ethics committee. Also, a flowchart should be added.

A3. Following the reviewer's suggestions, we have highlighted the type of study in the methods section (Please see L339). The code of the certificate of the ethics committee is mentioned in L350 to L353. On the other hand, Figure S1 illustrates a flowchart of experimental design.

Reviewer 2 Report

In this manuscript, the authors report the effects of physical exercise on AD and its impact on amyloid angiopathy and NVU using a triple transgenic animal model for AD. The manuscript is very well-written and a pleasant read. The introduction is well cited, results are presented clearly, and discussion is logical.

In line 29, the “ACC” should be changed with CAA.

Overall, I think this article is acceptable for publication after this minor change.

Author Response

Reviewer 2

In this manuscript, the authors report the effects of physical exercise on AD and its impact on amyloid angiopathy and NVU using a triple transgenic animal model for AD. The manuscript is very well-written and a pleasant read. The introduction is well cited, results are presented clearly, and discussion is logical.

  • We appreciate the effort and dedication of the reviewer for analyzing our manuscript.

Q1. In line 29, the “ACC” should be changed with CAA.

A1. The change was made (highlighted in green).

Overall, I think this article is acceptable for publication after this minor change